# MeshFormer: High-Quality Mesh Generation with 3D-Guided Reconstruction Model

Minghua Liu[*,1,2,†]  Chong Zeng[*,3,‡]  Xinyue Wei[1,2,†]  Ruoxi Shi[1,2,†]
Linghao Chen[2,3,†]  Chao Xu[2,4,†]  Mengqi Zhang[2]  Zhaoning Wang[5]
Xiaoshuai Zhang[1,2,†]  Isabella Liu[1]  Hongzhi Wu[3]  Hao Su[1,2]

[1] UC San Diego  [2] Hillbot Inc.  [3] Zhejiang University  [4] UCLA  [5] University of Central Florida

Project Website: `https://meshformer3d.github.io/`

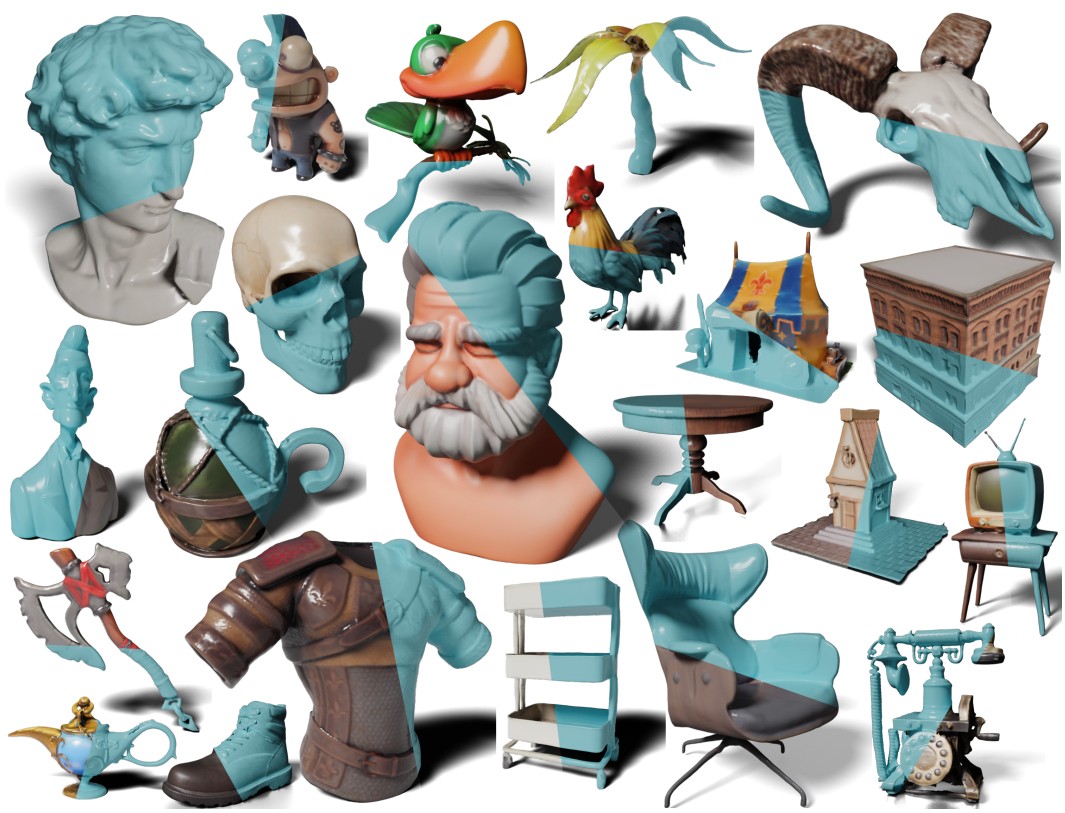

Figure 1: Given a sparse set (e.g., 6) of multi-view RGB images and their normal maps as input, MeshFormer reconstructs high-quality 3D textured meshes with fine-grained, sharp geometric details in a feed-forward pass of just a few seconds. Here, ground truth multi-view RGB and normal images are used as input.

## Abstract

Open-world 3D reconstruction models have recently garnered significant attention. However, without sufficient 3D inductive bias, existing methods typically entail expensive training costs and struggle to extract high-quality 3D meshes. In this

---

*  Equal contribution. † Work done during internship at Hillbot Inc. ‡ Work done during internship at UC San Diego.

38th Conference on Neural Information Processing Systems (NeurIPS 2024).

work, we introduce MeshFormer, a sparse-view reconstruction model that explicitly leverages 3D native structure, input guidance, and training supervision. Specifically, instead of using a triplane representation, we store features in 3D sparse voxels and combine transformers with 3D convolutions to leverage an explicit 3D structure and projective bias. In addition to sparse-view RGB input, we require the network to take input and generate corresponding normal maps. The input normal maps can be predicted by 2D diffusion models, significantly aiding in the guidance and refinement of the geometry's learning. Moreover, by combining Signed Distance Function (SDF) supervision with surface rendering, we directly learn to generate high-quality meshes without the need for complex multi-stage training processes. By incorporating these explicit 3D biases, MeshFormer can be trained efficiently and deliver high-quality textured meshes with fine-grained geometric details. It can also be integrated with 2D diffusion models to enable fast single-image-to-3D and text-to-3D tasks.

# 1    Introduction

High-quality 3D meshes are essential for numerous applications, including rendering, simulation, and 3D printing. Traditional photogrammetry systems [57, 61] and recent neural approaches, such as NeRF [43], typically require a dense set of input views of the object and long processing times. Recently, open-world 3D object generation has made significant advancements, aiming to democratize 3D asset creation by reducing input requirements. There are several prevailing paradigms: training a native 3D generative model using only 3D data [13, 95] or performing per-shape optimization with Score Distillation Sampling (SDS) losses [30, 47]. Another promising direction is to first predict a sparse set of multi-view images using 2D diffusion models [33, 59] and then lift these predicted images into a 3D model by training a feed-forward network [31, 32]. This strategy addresses the limited generalizability of models trained solely on 3D data and overcomes the long runtime and 3D inconsistency of per-shape-optimization-based methods.

While many recent works explore utilizing priors from 2D diffusion models, such as generating consistent multi-view images [59, 60] and predicting normal maps from RGB [12, 37, 59], the feed-forward model that converts multi-view images into 3D remains underexplored. One-2-3-45 [32] leverages a generalizable NeRF method for 3D reconstruction but suffers from limited quality and success rates. One-2-3-45++ [31] improves on this by using a two-stage 3D diffusion model, yet it still struggles to generate high-quality textures or fine-grained geometry. Given that sparse-view reconstruction of open-world objects requires extensive priors, another family of works pioneered by the large reconstruction model (LRM) [16] combines large-scale transformer models with the triplane representation and trains the model primarily using rendering loss. Although straightforward, these methods typically require over a hundred GPUs to train. Moreover, due to their reliance on volume rendering, these methods have difficulty extracting high-quality meshes. For instance, some recent follow-up works [79, 85] implement complex multi-stage "NeRF-to-mesh" training strategies, but the results still leave room for improvement.

In this work, we present MeshFormer, an open-world sparse-view reconstruction model that takes a sparse set of posed images of an arbitrary object as input and delivers high-quality 3D textured meshes with a single forward pass in a few seconds. Instead of representing 3D data as "2D planes" and training a "black box" transformer model optimizing only rendering loss, we find that by incorporating various types of 3D-native priors into the model design, including network architecture, supervision signals, and input guidance, our model can significantly improve both mesh quality and training efficiency. Specifically, we propose representing features in explicit 3D voxels and introduce a novel architecture that combines large-scale transformers with 3D (sparse) convolutions. Compared to triplanes and pure transformers models with little 3D-native design, MeshFormer leverages the explicit 3D structure of voxel features and the precise projective correspondence between 3D voxels and 2D multi-view features, enabling faster and more effective learning.

Unlike previous works that rely on NeRF-based representation in their pipeline, we utilize mesh representation throughout the process and train MeshFormer in a unified, single-stage manner. Specifically, we propose combining surface rendering with additional explicit 3D supervision, requiring the model to learn a signed distance function (SDF) field. The network is trained with high-resolution SDF supervision, and efficient differentiable surface rendering is applied to the extracted meshes

for rendering losses. Due to the explicit 3D geometry supervision, MeshFormer enables faster training while eliminating the need for expensive volume rendering and learning an initial coarse NeRF. Furthermore, in addition to multi-view posed RGB images, we propose using corresponding normal maps as input, which can be captured through sensors and photometric techniques [4, 82] or directly estimated by recent 2D vision models [12, 37, 59]. These multi-view normal images provide important clues for 3D reconstruction and fine-grained geometric details. We also task the model with learning a normal texture in addition to the RGB texture, which can then be used to enhance the generated geometry through a traditional post-processing algorithm [44].

Thanks to the explicit 3D-native structure, supervision signal, and normal guidance that we have incorporated, MeshFormer can generate high-quality textured meshes with fine-grained geometric details, as shown in Figure 1. Compared to concurrent methods that require over one hundred GPUs or complex multi-stage training, MeshFormer can be trained more efficiently and conveniently with just eight GPUs over two days, achieving on-par or even better performance. It can also seamlessly integrate with various 2D diffusion models to enable numerous tasks, such as single-image-to-3D and text-to-3D. In summary, our key contributions include:

- We introduce MeshFormer, an open-world sparse-view reconstruction model capable of generating high-quality 3D textured meshes with fine-grained geometric details in a few seconds. It can be trained with only 8 GPUs, outperforming baselines that require over one hundred GPUs.

- We propose a novel architecture that combines 3D (sparse) convolution and transformers. By explicitly leveraging 3D structure and projective bias, it facilitates better and faster learning.

- We propose a unified single-stage training strategy for generating high-quality meshes by combining surface rendering and explicit 3D geometric supervision.

- We are the first to introduce multi-view normal images as input to the feed-forward reconstruction network, providing crucial geometric guidance. Additionally, we propose to predict extra 3D normal texture for geometric enhancement.

## 2 Related Work

**Open-world 3D Object Generation** Thanks to the emergence of large-scale 3D datasets [8, 9] and the extensive priors learned by 2D models [50, 51, 55, 56], open-world 3D object generation have recently made significant advancements. Exemplified by DreamFusion [47], a line of work [5, 6, 10, 26, 30, 48, 58, 60, 62, 65, 70, 76] uses 2D models as guidance to generate 3D objects through per-shape optimization with SDS-like losses. Although these methods produce increasingly better results, they are still limited by lengthy runtimes and many other issues. Another line of work [16, 20, 40, 45, 84, 96] trains a feed-forward generative model solely on 3D data that consumes text prompts or single-image inputs. While fast during inference, these methods struggle to generalize to unseen object categories due to the scarcity of 3D data. More recently, works such as Zero123 [33] have shown that 2D diffusion models can be fine-tuned with 3D data for novel view synthesis. A line of work [27, 27, 31, 64, 77, 79, 85], pioneered by One-2-3-45 [32], proposes first predicting multi-view images through 2D diffusion models and then lifting them to 3D through a feed-forward network, effectively addressing the speed and generalizability issues. Many recent works have also explored better strategies to fine-tune 2D diffusion models for enhancing the 3D consistency of multi-view images [14, 17, 23, 34, 36, 49, 59, 60, 69, 72, 80, 81, 89, 91]. In addition to the feed-forward models, the generated multi-view images can also be lifted to 3D through optimizations [14, 34, 37].

**Sparse-View Feed-Forward Reconstruction Models** When a small baseline between input images is assumed, existing generalizable NeRF methods [35, 52, 68, 88] aim to find pixel correspondences and learn generalizable priors across scenes by leveraging cost-volume-based techniques [3, 38, 90] or transformer-based structures [19, 24, 54, 71, 74]. Some of methods have also incorporated a 2D diffusion process into the pipeline [1, 21, 66]. However, these methods often struggle to handle large baseline settings (e.g., only frontal-view reconstruction) or are limited by a small training set and fail to generalize to open-world objects. Recently, many models [27, 64, 73, 77, 79, 85–87, 92, 94] specifically aimed at open-world 3D object generation have been proposed. They typically build large networks and aim to learn extensive reconstruction priors by training on large-scale 3D datasets [9]. For example, the triplane representation and transformer models are often used. By applying volume rendering or Gaussian splatting [64, 86, 92], they train the model with rendering losses. However, these methods typically require extensive GPUs to train and have difficulty extracting high-quality

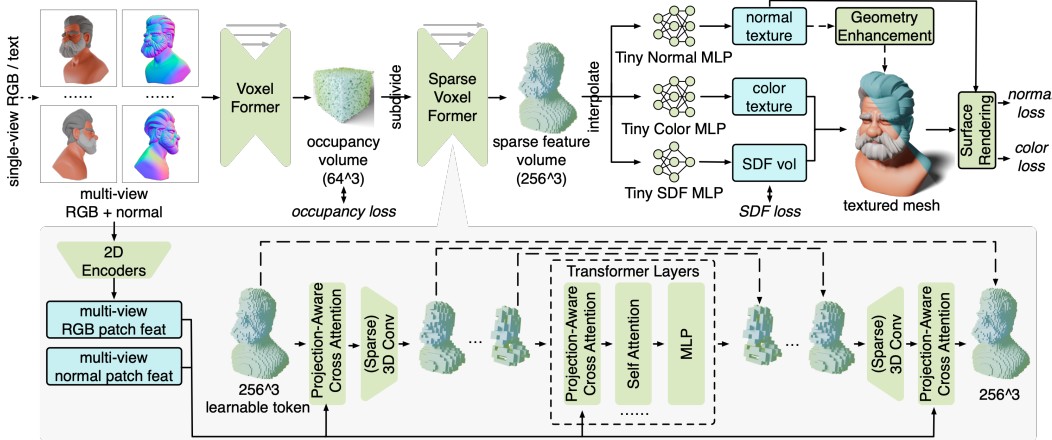

Figure 2: **Pipeline Overview.** MeshFormer takes a sparse set of multi-view RGB and normal images as input, which can be estimated using existing 2D diffusion models. We utilize a 3D feature volume representation, and submodules Voxel Former and Sparse Voxel Former share a similar novel architecture, detailed in the gray region. We train MeshFormer in a unified single stage by combining mesh surface rendering and $512^3$ SDF supervision. MeshFormer learns an additional normal texture, which can be used to further enhance the geometry and generate fine-grained sharp geometric details.

meshes. While some recent (concurrent) works [79, 85] utilize multi-stage "NeRF-to-mesh" training strategies to improve the quality, the results still leave room for improvement.

**Geometry Guidance for 3D Reconstruction** Many recent works have shown that in addition to multi-view RGB images, 2D diffusion models can be fine-tuned to generate other geometric modalities, such as depth maps [75], normal maps [12, 37, 41], or coordinate maps [28, 77]. These additional modalities can provide crucial guidance for 3D generation and reconstruction. While many recent methods utilize these geometric cues as inverse optimization guidance [5, 12, 28, 37, 49, 77], we propose to take normal maps as input in a feed-forward reconstruction model and task the model with generating 3D-consistent normal texture for geometry enhancement of sharp details.

**3D Native Representations and Network Architectures in 3D Generation** The use of 3D voxel representations and 3D convolutions is common in general 3D generation. However, most recent works focus on 3D-native diffusion [7, 18, 29, 31, 53, 95], one of the key paradigms in 3D generation, which differs from the route taken by MeshFormer. These 3D-diffusion-based methods have some common limitations. For instance, they focus solely on geometry generation and cannot directly predict high-quality textures from the network [7, 18, 29, 31, 53, 95]. Due to the limited availability of 3D data, 3D-native diffusion methods also typically struggle with open-world capabilities and are often constrained to closed-domain datasets (e.g., ShapeNet [2]) in their experiments [7, 29, 95].

In MeshFormer, our goal is to achieve direct high-quality texture generation while handling arbitrary object categories. Therefore, we adopt a different approach: sparse-view feed-forward reconstruction, as opposed to 3D-native diffusion. In this specific task setting, more comparable works are recent LRM-style methods [64, 67, 79, 85]. However, most of these methods rely on a combination of triplane representation and large-scale transformers. In this paper, we demonstrate that 3D-native representations and networks can not only be used in 3D-native diffusion but can also be combined with differentiable rendering to train a feed-forward sparse-view reconstruction model using rendering losses. In open-world sparse-view reconstruction, we are not limited to the triplane representation. Instead, 3D-native structures (e.g., voxels), network architectures, and projective priors can facilitate more efficient training, significantly reducing the required training resources. While scalable networks are necessary to learn extensive priors, scalability is not exclusive to triplane-based transformers. By integrating 3D convolutions with transformer layers, scalability can also be achieved.

## 3 Method

As shown in Figure 2, MeshFormer takes a sparse set of posed multi-view RGB and normal images as input and generates a high-quality textured mesh in a single feed-forward pass. In the following

sections, we will first introduce our choice of 3D representation and a novel model architecture that combines large-scale transformers with 3D convolutions (Sec. 3.1). Then, we will describe our training objectives, which integrate surface rendering and explicit 3D SDF supervision (Sec. 3.2). Last but not least, we will present our normal guidance and geometry enhancement module, which plays a crucial role in generating high-quality meshes with fine-grained geometric details (Sec. 3.3).

## 3.1 3D Representation and Model Architecture

**Triplane vs. 3D Voxels** Open-world sparse-view reconstruction requires extensive priors, which can be learned through a large-scale transformer. Prior arts [27, 67, 77, 79, 85] typically utilize the triplane representation, which decomposes a 3D neural field into a set of 2D planes. While straightforward for processing by transformers, the triplane representation lacks explicit 3D spatial structures and makes it hard to enable precise interaction between each 3D location and its corresponding 2D projected pixels from multi-view images. For instance, these methods often simply apply self-attention across all triplane patch tokens and cross-attention between triplane tokens and all multi-view image tokens. This all-to-all attention is not only costly but also makes the methods cumbersome to train. Moreover, the triplane representation often shows results with notable artifacts at the boundaries of patches and may suffer from limited expressiveness for complex structures. Consequently, we choose the 3D voxel representation instead, which explicitly preserves the 3D spatial structures.

**Combining Transformer with 3D Convolution** To leverage the explicit 3D structure and the powerful expressiveness of a large-scale transformer model while avoiding an explosion of computational costs, we propose VoxelFormer and SparseVoxelFormer, which follow a 3D UNet architecture while integrating a transformer at the bottleneck. The overall idea is that we use local 3D convolution to encode and decode a high-resolution 3D feature volume, while the global transformer layer handles reasoning and memorizing priors for the compressed low-resolution feature volume. Specifically, as shown in Figure 2, a 3D feature volume begins with a learnable token shared by all 3D voxels. With the 3D voxel coordinates, we can leverage the projection matrix to enable each 3D voxel to aggregate 2D local features from multi-view images via a projection-aware cross-attention layer. By iteratively performing projection-aware cross-attention and 3D (sparse) convolution, we can compress the 3D volume to a lower-resolution one. After compression, each 3D voxel feature then serves as a latent token, and a deep transformer model is applied to a sequence of all 3D voxel features (position encoded) to enhance the model's expressiveness. Finally, we use the convolution-based inverse upper branch with skip connection to decode a 3D feature volume with the initial high resolution.

**Projection-Aware Cross Attention** Regarding 3D-2D interaction, the input multi-view RGB and normal images are initially processed by a 2D feature extractor, such as a trainable DINOv2 [46], to generate multi-view patch features. While previous cost-volume-based methods [3, 38] typically use mean or max pooling to aggregate multi-view 2D features, these simple pooling operations might be suboptimal for addressing occlusion and visibility issues. Instead, we propose a projection-aware cross-attention mechanism to adaptively aggregate the multi-view features for each 3D voxel. Specifically, we project each 3D voxel onto the $m$ views to interpolate $m$ RGB and normal features. We then concatenate these local patch features with the projected RGB and normal values to form $m$ 2D features. In the projection-aware cross-attention module, we use the 3D voxel feature to calculate a query and use both the 3D voxel feature and the $m$ 2D features to calculate $m + 1$ keys and values. A cross-attention is then performed for each 3D voxel, enabling precise interaction between each 3D location and its corresponding 2D projected pixels, and allowing adaptive aggregation of 2D features, which can be formulated as:

$$v \leftarrow \text{CrossAttention}(Q = \{v\}, K = \{p_i^v\}_{i=1}^m + \{v\}, V = \{p_i^v\}_{i=1}^m + \{v\}) \tag{1}$$

Where $v$ denotes a 3D voxel feature, and $p_i^v$ denotes its projected 2D pixel feature from view $i$, which is a concatenation of the RGB feature $f_i^v$, the normal feature $g_i^v$, and the RGB and normal values $c_i^v$ and $n_i^v$, respectively.

**Coarse-to-Fine Feature Generation** As shown in Fig. 2, to generate a high-resolution 3D feature volume that captures the fine-grained details of 3D shapes, we follow previous work [31, 95] by employing a coarse-to-fine strategy. Specifically, we first use VoxelFormer, which is equipped with full 3D convolution, to predict a low-resolution (e.g., $64^3$), coarse 3D occupancy volume. Each voxel in this volume stores a binary value indicating whether it is close to the surface. The predicted occupied voxels are then subdivided to create higher-resolution sparse voxels (e.g., $256^3$). Next, we utilize a second module, SparseVoxelFormer, which features 3D sparse convolution [63], to predict features for these sparse voxels. After this, we trilinearly interpolate the 3D feature of any

near-surface 3D point, which encodes both geometric and color information, from the high-resolution sparse feature volume. The features are then fed into various MLPs to learn the corresponding fields.

## 3.2 Unified Single-Stage Training: Surface Rendering with SDF Supervision

Existing works typically use NeRF [42] and volume rendering or 3D Gaussian splatting [22] since they come with a relatively easy and stable learning process. However, extracting high-quality meshes from their results is often non-trivial. For example, directly applying Marching Cubes [39] to density fields of learned NeRFs typically generates meshes with many artifacts. Recent methods [78, 79, 85] have designed complex, multi-stage "NeRF-to-mesh" training with differentiable surface rendering, but the generated meshes still leave room for improvement. On the other hand, skipping a good initialization and directly learning meshes from scratch using purely differentiable surface rendering losses is also infeasible, as it is highly unstable to train and typically results in distorted geometry.

In this work, we propose leveraging explicit 3D supervision in addition to 2D rendering losses. As shown in Figure 2, we task MeshFormer with learning a signed distance function (SDF) field supervised by a high-resolution (e.g., $512^3$) ground truth SDF volume. The SDF loss provides explicit guidance for the underlying 3D geometry and facilitates faster learning. It also allows us to use mesh representation and differentiable surface rendering from the beginning without worrying about good geometry initialization or unstable training, as the SDF loss serves as a strong regularization for the underlying geometry. By combining surface rendering with explicit 3D SDF supervision, we train MeshFormer in a unified, single-stage training process. As shown in Figure 2, we employ three tiny MLPs that take as input the 3D feature interpolated from the 3D sparse feature volume to learn an SDF field, a 3D color texture, and a 3D normal texture. We extract meshes from the SDF volume using dual Marching Cubes [39] and employ NVDiffRast [25] for differentiable surface rendering. We render both the multi-view RGB and normal images and compute the rendering losses, which consist of both the MSE and perceptual loss terms. As a result, our training loss can be expressed as:

$$\mathcal{L} = \lambda_1 \mathcal{L}_{\text{MSE}}^{\text{color}} + \lambda_2 \mathcal{L}_{\text{LPIPS}}^{\text{color}} + \lambda_3 \mathcal{L}_{\text{MSE}}^{\text{normal}} + \lambda_4 \mathcal{L}_{\text{LPIPS}}^{\text{normal}} + \lambda_5 \mathcal{L}_{\text{occ}} + \lambda_6 \mathcal{L}_{\text{SDF}} \quad (2)$$

where $L_{\text{occ}}$ and $L_{\text{SDF}}$ are MSE losses for occupancy and SDF volumes, and $\lambda_i$ denotes the weight of each loss term. Note that we do not use mesh geometry to derive normal maps; instead, we utilize the learned normal texture from the MLP, which will be detailed later.

## 3.3 Fine-Grained Geometric Details: Normal Guidance and Geometry Enhancement

Without dense-view correspondences, 3D reconstruction from sparse-view RGB images typically struggles to capture geometric details and suffers from texture ambiguity. While many recent works [27, 79, 85] attempt to employ large-scale models to learn mappings from RGB to geometric details, this typically requires significant computational resources. Additionally, these methods are primarily trained using 3D data, but it's still uncertain whether the scale of 3D datasets is sufficient for learning such extensive priors. On the other hand, unlike RGB images, normal maps explicitly encode geometric information and can provide crucial guidance for 3D reconstruction. Notably, open-world normal map estimation has achieved great advancements. Many recent works [12, 37, 59] demonstrate that 2D diffusion models, trained on billions of natural images, embed extensive priors and can be fine-tuned to predict normal maps. Given the significant disparity in data scale between 2D and 3D datasets, it may be more effective to use 2D models first for generating geometric guidance.

**Input Normal Guidance** As shown in Figure 2, in addition to multi-view RGB images, MeshFormer also takes multi-view normal maps as input, which can be generated using recent open-world normal estimation models [12, 37, 59]. In our experiments, we utilize Zero123++ v1.2 [59], which trains an additional ControlNet [93] over the multi-view prediction model. The ControlNet takes multi-view RGB images, predicted by Zero123++, as a condition and produces corresponding multi-view normal maps, expressed in the camera coordinate frame. Given these maps, MeshFormer first converts them to a unified world coordinate frame, and then treats them similarly to the multi-view RGB images, using projection-aware cross-attention to guide 3D reconstruction. According to our experiments (Sec. 4.4), the multi-view normal maps enable the networks to better capture geometry details, and thus greatly improve final mesh quality.

**Geometry Enhancement** While the straightforward approach of deriving normal maps from the learned mesh and using a normal loss to guide geometry learning has been commonly used, we find that this approach makes our mesh learning less stable. Instead, we propose learning a 3D normal texture, similar to a color texture, using a separate MLP. By computing the normal loss for MLP-queried normal maps instead of mesh-derived normal maps, we decouple normal texture learning

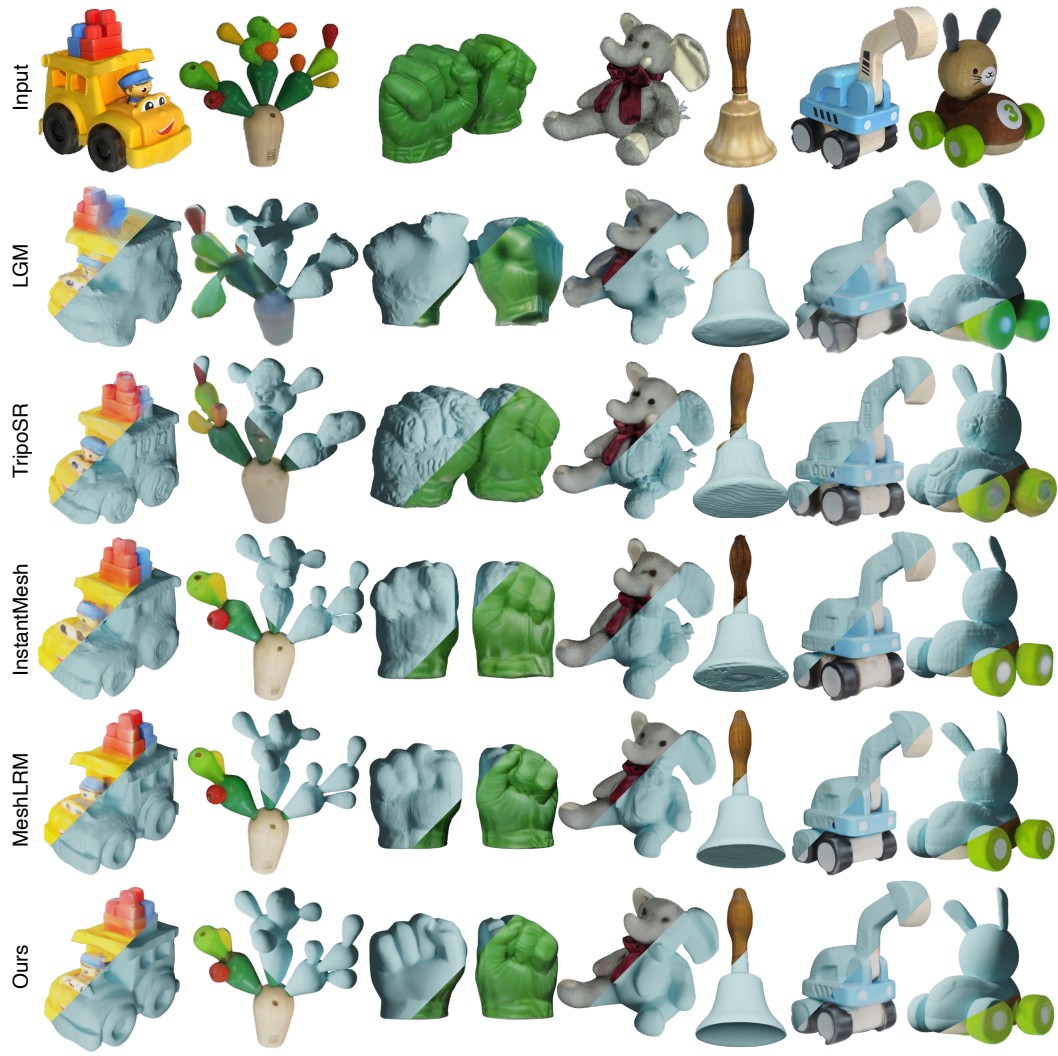

Figure 3: **Qualitative Examples of Single Image to 3D (GSO dataset).** Both the textured and textureless mesh renderings are shown. Please zoom in to examine details and mesh quality, and refer to the supplemental material for results of One-2-3-45++ [31] and CRM [77].

from underlying geometry learning. This makes the training more stable, as it is easier to learn a sharp 3D normal map than to directly learn a sharp mesh geometry. The learned 3D normal texture can be exported with the mesh, similar to the color texture, to support various graphics rendering pipelines. In applications that require precise 3D geometry, such as 3D printing, the learned normal texture can also be used to refine the mesh geometry with traditional algorithms. Specifically, during inference, after extracting a 3D mesh from the SDF volume, we utilize a post-processing algorithm [44] that takes as input the 3D positions of the mesh vertices and the vertex normals estimated from the MLP. The algorithm adjusts the mesh vertices to align with the predicted normals in a few seconds, further enhancing the geometry quality and generating sharp geometric details, as shown in Figure 5.

## 4 Experiments

### 4.1 Implementation Details and Evaluation Settings

**Implementation Details** We trained MeshFormer on the Objaverse [9] dataset. The total number of network parameters is approximately 648 million. We trained the model using 8 H100 GPUs for about one week (350k iterations) with a batch size of 1 per GPU, although we also show that the model can achieve similar results in just two days. Please refer to the supplementary for more details.

Table 1: **Quantitative Results of Single Image to 3D.** Evaluated on the 1,030 and 1,038 3D shapes from the GSO [11] and the OmniObject3D [83] datasets, respectively. One-2-3-45++ [31], InstantMesh [85], MeshLRM [79], and our method all take the same multi-view RGB images predicted by Zero123++ [59] as input. CD denotes Chamfer Distance.

| Method | GSO [11] | | | | OmniObject3D [83] | | | |
|---|---|---|---|---|---|---|---|---|
| | F-Score ↑ | CD ↓ | PSNR ↑ | LPIPS ↓ | F-Score ↑ | CD ↓ | PSNR ↑ | LPIPS ↓ |
| One-2-3-45++ [31] | 0.936 | 0.039 | 20.97 | 0.21 | 0.871 | 0.054 | 17.08 | 0.31 |
| TripoSR [67] | 0.896 | 0.047 | 19.85 | 0.26 | 0.895 | 0.048 | 17.68 | 0.28 |
| CRM [77] | 0.886 | 0.051 | 19.99 | 0.27 | 0.821 | 0.065 | 16.01 | 0.34 |
| LGM [64] | 0.776 | 0.074 | 18.52 | 0.35 | 0.635 | 0.114 | 14.75 | 0.45 |
| InstantMesh [64] | 0.934 | 0.037 | 20.90 | 0.22 | 0.889 | 0.049 | 17.61 | 0.28 |
| MeshLRM [79] | 0.956 | 0.033 | 21.31 | **0.19** | 0.910 | 0.045 | 18.10 | **0.26** |
| Ours | **0.963** | **0.031** | **21.47** | 0.20 | **0.914** | **0.043** | **18.14** | 0.27 |

**Evaluation Settings** We evaluate the methods on two datasets: GSO [11] and OmniObject3D [83]. Both datasets contain real-scanned 3D objects that were not seen during training. For the GSO dataset, we use all 1,030 3D shapes for evaluation. For the OmniObject3D dataset, we randomly sample up to 5 shapes from each category, resulting in 1,038 shapes for evaluation. We utilize both 2D and 3D metrics. For 3D metrics, we use both the F-score and Chamfer distance (CD), calculated between the predicted meshes and ground truth meshes, following [31, 85]. For 2D metrics, we compute both PSNR and LPIPS for the rendered color images. Since each baseline may use a different coordinate frame for generated results, we carefully align the predicted meshes of all methods to the ground truth meshes before calculating the metrics. Please refer to the supplemental material for more details.

### 4.2 Comparison with Single/Sparse-View to 3D Methods

We compare MeshFormer with recent open-world feed-forward single/sparse-view to 3D methods, including One-2-3-45++ [31], TripoSR [67], CRM [77], LGM [64], InstantMesh [85], and MeshLRM [79]. Many of these methods have been released recently and should be considered concurrent methods. For MeshLRM [79], we contacted the authors for the results. For the other methods, we utilized their official implementations. Please refer to the supplementary for details.

Since input settings differ among the baselines, we evaluate all methods in a unified single-view to 3D setting. For the GSO dataset, we utilized the first thumbnail image as the single-view input. For the OmniObject3D dataset, we used a rendered image with a random pose as input. For One-2-3-45++ [31], InstantMesh [85], MeshLRM [79], and our MeshFormer, we first utilized Zero123++ [59] to convert the input single-view image into multi-view images before 3D reconstruction. Other baselines follow their original settings and take a single-view image directly as input. In addition to the RGB images, our MeshFormer also takes additional multi-view normal images as input, which are also predicted by Zero123++ [59]. **Note that when comparing with baseline methods, we never use ground truth normal images to ensure a fair comparison.**

In Fig. 3, we showcase qualitative examples. Our MeshFormer produces the most accurate meshes with fine-grained, sharp geometric details. In contrast, baseline methods produce inferior mesh quality. For example, TripoSR directly extracts meshes from the learned NeRF representation, resulting in significant artifacts. While InstantMesh and MeshLRM use mesh representation in their second stage, notable uneven artifacts are still observable upon a zoom-in inspection. Additionally, all baseline methods incorrectly close the surface of the copper bell. We also provide quantitative results in Tab. 1. Although our baselines include four methods released just one or two months before the time of submission, our MeshFormer significantly outperforms many of them and achieves the best performance on most metrics across two datasets. For the color LPIPS metric, our performance is very similar to MeshLRM's, despite a perceptual loss being their main training loss term. We also highlight that many of the baselines require over one hundred GPUs for training, whereas our model can be efficiently trained with just 8 GPUs. Please refer to Sec. 4.4 for analysis on training efficiency.

### 4.3 Application: Text to 3D

In addition to the single image to 3D, MeshFormer can also be integrated with 2D diffusion models to enable various 3D object generation tasks. For example, we follow the framework proposed by [37] to finetune Stable Diffusion [56] and build a text-to-multi-view model. By integrating this

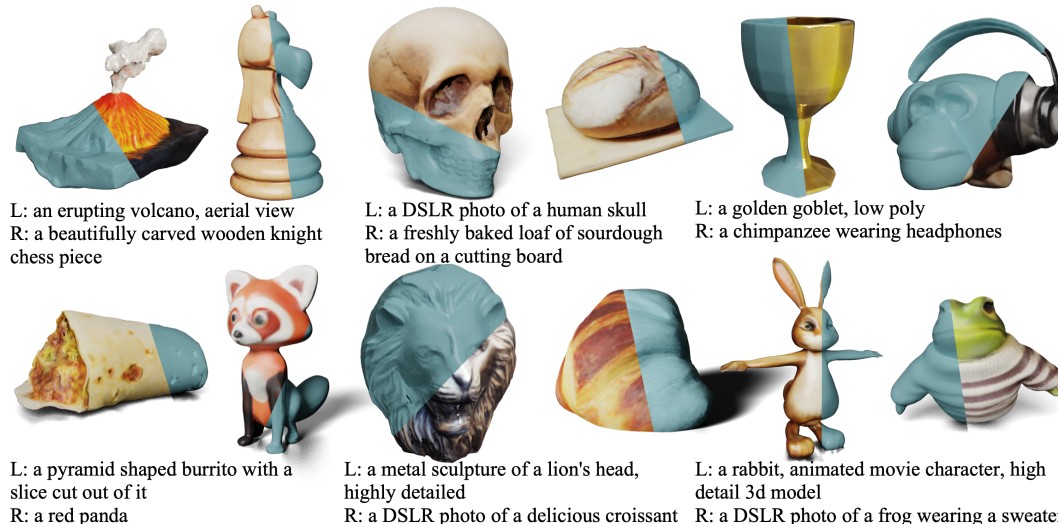

L: an erupting volcano, aerial view
R: a beautifully carved wooden knight chess piece

L: a DSLR photo of a human skull
R: a freshly baked loaf of sourdough bread on a cutting board

L: a golden goblet, low poly
R: a chimpanzee wearing headphones

L: a pyramid shaped burrito with a slice cut out of it
R: a red panda

L: a metal sculpture of a lion's head, highly detailed
R: a DSLR photo of a delicious croissant

L: a rabbit, animated movie character, high detail 3d model
R: a DSLR photo of a frog wearing a sweater

Figure 4: **Application: Text to 3D.** Given a text prompt, a 2D diffusion model first predicts multi-view RGB and normal images, which are then fed to MeshFormer for 3D reconstruction. Please refer to the supplementary for comparisons with Instant3D [27].

Table 2: We compare methods using limited training resources. Evaluated on the GSO [11] dataset.

| Method | Training Resources | F-Score ↑ | CD ↓ | PSNR-C ↑ | LPIPS-C ↓ | PSNR-N ↑ | LPIPS-N ↓ |
|---|---|---|---|---|---|---|---|
| MeshLRM [79] | 8×H100 48h | 0.925 | 0.0397 | 21.09 | 0.26 | 21.69 | 0.22 |
| Ours | | 0.960 | 0.0317 | 21.41 | 0.20 | 23.01 | 0.15 |

model, along with the normal prediction from Zero123++ [59], with MeshFormer, we can enable the task of text to 3D. Figure 4 shows some interesting results, where we convert a single text prompt into a high-quality 3D mesh in just a few seconds. Please refer to the supplemental materials for a qualitative comparison with one of the state-of-the-art text-to-3D methods, Instant3D [27].

### 4.4 Analysis and Ablation Study

**Explicit 3D structure vs. Triplane** In Section 4.2, we demonstrated that MeshFormer outperforms baseline methods that primarily utilize the triplane representation. Here, we highlight two additional advantages of using the explicit 3D voxel structure: training efficiency and the avoidance of "triplane artifacts". Without leveraging explicit 3D structure, existing triplane-based large reconstruction models require extensive computing resources for training. For example, TripoSR requires 176 A100 GPUs for five days of training. InstantMesh relies on OpenLRM [15], which requires 128 A100 GPUs for three days of training. MeshLRM also utilizes similar resources during training. By utilizing explicit 3D structure and projective bias, our MeshFormer can be trained much more efficiently using only 8 GPUs. To better understand the gap, we trained both MeshLRM and our MeshFormer under very limited training resources, and the results are shown in Table 2. When using only 8 GPUs for two days, we found that MeshLRM failed to converge and experienced significant performance degradation compared to the results shown in Table 1, while our MeshFormer had already converged to a decent result, close to the fully-trained version, demonstrating superior training efficiency.

We observe that the triplane typically generates results with axis-aligned artifacts, as shown in Fig.3 (5th row, please zoom in). As demonstrated in the supplementary (Fig. 7), these artifacts also cause difficulties for MeshLRM [79] in capturing the words on objects. These limitations are likely caused by the limited number of triplane tokens (e.g., $32 \times 32 \times 3$), constrained by the global attention, which often leads to artifacts at the boundaries of the triplane patches. In contrast, MeshFormer leverages sparse voxels, supports a higher feature resolution of $256^3$, and is free from such artifacts.

**Normal Input and SDF supervision** As shown in Table 3 (a), the performance significantly drops when multi-view input normal maps are removed, indicating that the geometric guidance and clues provided by normal images are crucial for facilitating network training, particularly for local geometric details. In (f), we replace ground truth normal maps with normal predictions by Zero123++ [59] and

Table 3: **Ablation Study on the GSO [11] dataset.** -C denotes color renderings, and -N denotes normal renderings. CD stands for Chamfer distance. By default, ground truth multi-view images are used to exclude the influence of errors from 2D diffusion models.

|   | Setting | PSNR-C ↑ | LPIPS-C ↓ | PSNR-N ↑ | LPIPS-N ↓ | F-Score ↑ | CD ↓ |
|---|---|---|---|---|---|---|---|
| a | w/o normal input | 24.82 | 0.129 | 24.85 | 0.107 | 0.964 | 0.024 |
| b | w/o SDF supervision | 20.72 | 0.244 | 20.42 | 0.257 | 0.940 | 0.035 |
| c | w/o transformer layer | 26.63 | 0.101 | 29.80 | 0.036 | 0.992 | 0.013 |
| d | w/o projection-aware cross-attention | 25.48 | 0.155 | 29.01 | 0.045 | 0.991 | 0.013 |
| e | w/o geometry enhancement | 27.95 | 0.085 | 29.10 | 0.048 | 0.992 | 0.012 |
| f | w/ pred normal | 26.84 | 0.096 | 26.99 | 0.067 | 0.987 | 0.017 |
| g | full | 28.15 | 0.083 | 29.80 | 0.036 | 0.992 | 0.012 |

observe a notable performance gap compared to (g). This indicates that although predicted multi-view normal images can be beneficial, existing 2D diffusion models still have room for improvement in generating more accurate results. See supplementary for qualitative examples. As shown in (b), if we remove the SDF loss after the first epoch and train the network using only surface rendering losses, the geometry learning quickly deteriorates, resulting in poor geometry. This explains why existing methods [27, 79] typically employ complex multi-stage training and use volume rendering to learn a coarse NeRF in the initial stage. By leveraging explicit 3D SDF supervision as strong geometric regularization, we enable a unified single-stage training, using mesh as the only representation.

**Projection-Aware Cross-Attention and Transformer Layers** We propose to utilize projection-aware cross-attention to precisely aggregate multi-view projected 2D features for each 3D voxel. In conventional learning-based multi-view stereo (MVS) methods [3, 38], average or max pooling is typically employed for feature aggregation. In Table 3 (d), we replace the cross-attention with a simple average pooling and we observe a significant performance drop. This verifies that projection-aware cross-attention provides a more effective way for 3D-2D interaction while simple average pooling may fail to handle the occlusion and visibility issues. In the bottleneck of the UNet, we treat all 3D (sparse) voxels as a sequence of tokens and apply transformer layers to them. As shown in row (c), after removing these layers, we observe a performance drop in metrics related to texture quality. This indicates that texture learning requires more extensive priors and benefits more from the transformer layers.

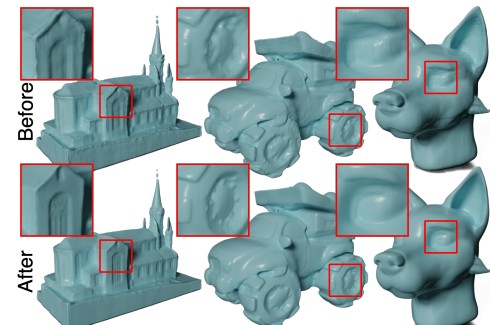

Figure 5: Geometry enhancement generates sharper details. Please zoom in to see the details.

**Geometry Enhancement** We propose to learn an additional normal map texture and apply a traditional algorithm as post-processing for geometry enhancement during inference. As shown in Figure 5, the geometry enhancement aligns the mesh geometry with the learned normal texture and generates fine-grained sharp details. In some cases (such as the wolf), the meshes output by the network are already good enough, and the difference caused by the enhancement tends to be subtle. Row (e) also quantitatively verifies the effectiveness of the module.

## 5 Conclusion and Limitations

We present MeshFormer, an open-world sparse-view reconstruction model that leverages explicit 3D native structure, supervision signals, and input guidance. MeshFormer can be conveniently trained in a unified single-stage manner and efficiently with just 8 GPUs. It generates high-quality meshes with fine-grained geometric details and outperforms baselines trained with over one hundred GPUs.

MeshFormer relies on 2D models to generate multi-view RGB and normal images from a single input image or text prompt. However, existing models still have limited capabilities to generate consistent multi-view images, which can cause a performance drop. Strategies to improve model robustness against such imperfect predictions are worth further exploration, and we leave this as future work.

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

# A    Appendix

## A.1    Comparison with Instant3D

In Figure 6, we showcase the comparison with Instant3D [27] on the text-to-3D task. The results are obtained from the paper authors. While Instant3D [27] also generates 3D shapes that match the input text prompt, our method generates results with superior mesh quality and fine-grained, sharp geometric details.

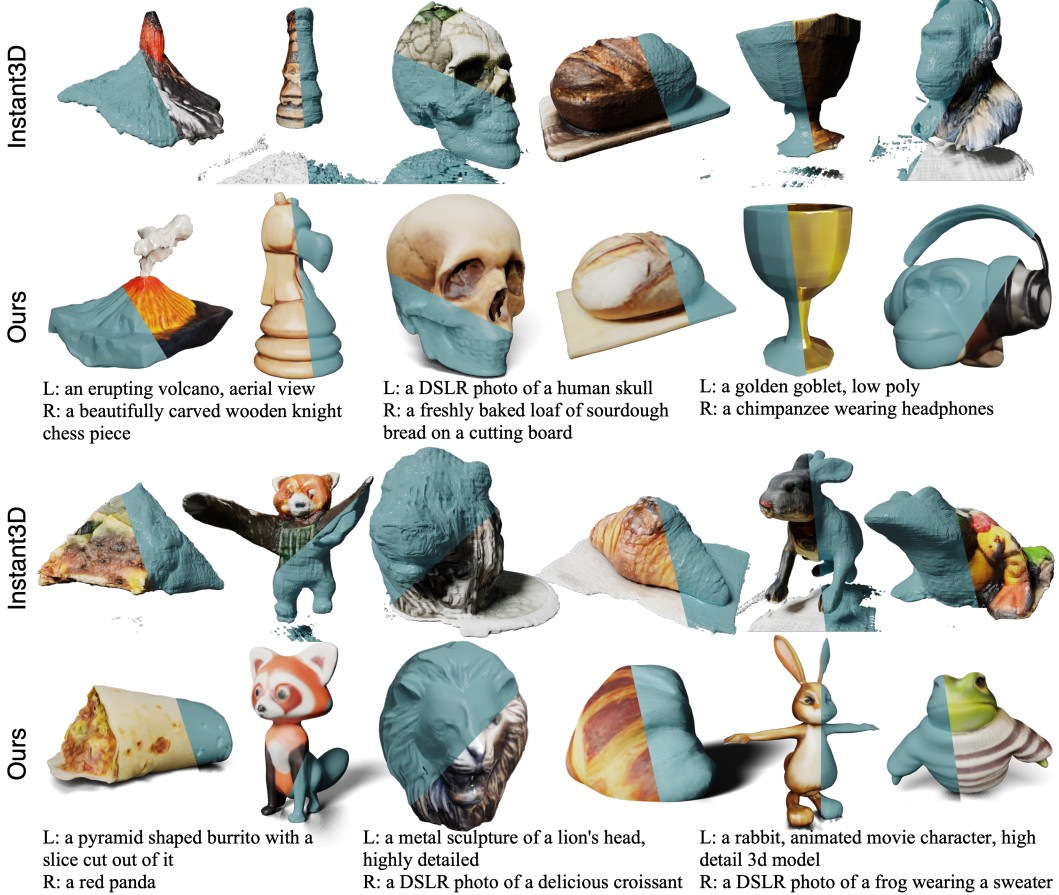

L: an erupting volcano, aerial view
R: a beautifully carved wooden knight chess piece

L: a DSLR photo of a human skull
R: a freshly baked loaf of sourdough bread on a cutting board

L: a golden goblet, low poly
R: a chimpanzee wearing headphones

L: a pyramid shaped burrito with a slice cut out of it
R: a red panda

L: a metal sculpture of a lion's head, highly detailed
R: a DSLR photo of a delicious croissant

L: a rabbit, animated movie character, high detail 3d model
R: a DSLR photo of a frog wearing a sweater

Figure 6: **Application: Text-to-3D**. Comparison with Instant3D [27].

## A.2    Triplane Artifacts

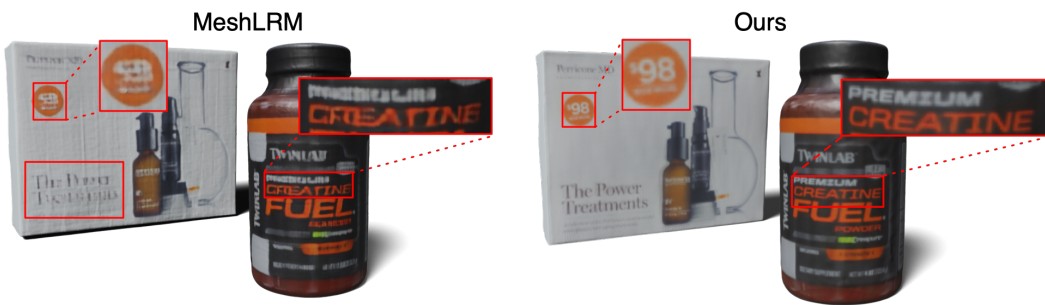

Figure 7: The triplane-based method MeshLRM [79] has difficulty capturing words on objects, even when ground truth multi-view RGB images are used as input.

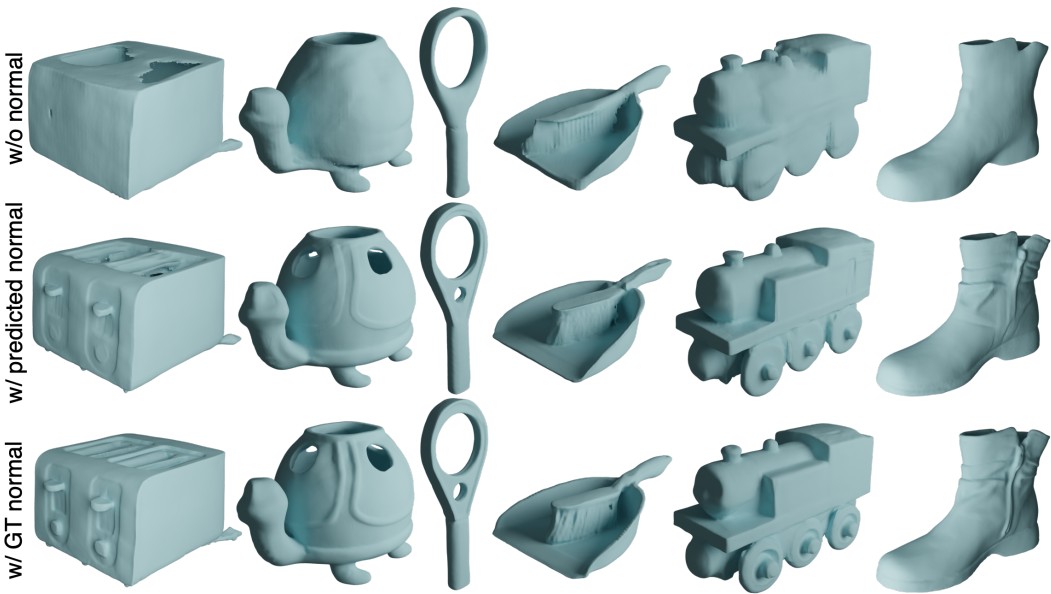

Figure 8: Ablation study on input normal maps. Evaluated on the GSO dataset [11]. "w/o normal" indicates that the model is trained with multi-view RGB images only. "w/ predicted normal" indicates that the model is trained with ground truth normal maps but evaluated with predicted normals by Zero123++ [59]. "w/ GT normal" indicates that the model is trained and tested with ground truth normals.

Table 4: Normal consistency (angle error) between the mesh geometry (mesh vertex normals) and the predicted normal maps, both before and after the geometry enhancement post-processing. The ratio of mesh vertices below a specific error threshold is shown. Evaluated on the GSO dataset.

| angle error threshold | before | after |
|---|---|---|
| $< 1°$ | 8.83% | 16.27% |
| $< 2°$ | 26.39% | 40.83% |
| $< 5°$ | 60.55% | 73.19% |
| $< 10°$ | 78.79% | 86.43% |
| $< 15°$ | 86.46% | 91.29% |

As shown in Fig.7, MeshLRM [79] has difficulty capturing words on objects, even when ground truth multi-view RGB images are used as input. We speculate that this is due to the limited number of triplane patches (e.g., $32 \times 32 \times 3$) restricted by global attention. In contrast, our method leverages sparse voxels and supports a much higher feature resolution of $256^3$, making it free from such issues.

### A.3 Ablation Study: Input Normal Maps

In Figure 8, we qualitatively demonstrate the effect of input normal maps. When the model is trained without multi-view normal maps, we find that the generated model can only capture the global 3D shape but fails to generate fine-grained geometric details. However, when the model is given predicted normal maps, the performance is significantly better, although there are still some small gaps when compared to the results of ground truth normals (see the bread hole of the toaster and the wheel of the tram). This indicates errors or inconsistencies from the 2D normal prediction models.

### A.4 Ablation Study: Geometry Enhancement

We propose asking the network to predict an additional normal texture, which can be used for further geometric enhancement by applying a traditional algorithm as post-processing. The geometric enhancement aims to align the mesh geometry with the predicted normal map by adjusting the vertex locations. However, the traditional algorithm we used cannot guarantee that the mesh normals will

Table 5: **Analysis of our mesh generation quality over training time.** Evaluated on the GSO [11] dataset.

| Training Time | PSNR-C ↑ | LPIPS-C ↓ | PSNR-N ↑ | LPIPS-N ↓ | CD ↓ | F-Score ↑ |
|---|---|---|---|---|---|---|
| 8×H100 12h | 21.28 | 0.2135 | 22.89 | 0.1536 | 0.0330 | 0.960 |
| 8×H100 24h | 21.32 | 0.2076 | 22.96 | 0.1516 | 0.0320 | 0.960 |
| 8×H100 48h | 21.41 | 0.2033 | 23.01 | 0.1484 | 0.0317 | 0.960 |
| 8×H100 120h | 21.44 | 0.2029 | 23.04 | 0.1480 | 0.0314 | 0.961 |
| 8×H100 168h | 21.47 | 0.2010 | 23.09 | 0.1466 | 0.0313 | 0.963 |

be fully aligned with the predicted normal maps after processing. This limitation arises because the algorithm operates in local space and avoids large vertex displacements. Moreover, the predicted normal maps may contain errors or inconsistencies, such as conflicting neighboring normals. The adopted algorithm is an iterative numerical optimization method and does not compute an analytic solution.

However, we have quantitatively verified that the post-processing module can significantly improve normal consistency with the predicted normal map. For example, before post-processing, only 26.4% of mesh vertices had a normal angle error of less than 2 degrees. After post-processing, this number increased to 40.8%. For a 10-degree threshold, the ratio increases from 78.8% to 86.4%. For more details, please refer to Table 4.

## A.5  Ablation Study: Training time

Our MeshFormer can be trained efficiently using only 8 GPUs, typically converging in approximately two days. Table 5 presents a quantitative analysis of our mesh generation quality over the training period. We observe that performance improves rapidly and nearly converges, with only marginal changes occurring after the two-day training period.

## A.6  Training Details and Evaluation Metrics

**Training Details:** We trained the model using a subset of 395k 3D shapes filtered from the Objaverse [9] dataset. These objects have a distributable Creative Commons license and were obtained by the Objaverse team using Sketchfab's public API. For each filtered 3D shape, we randomly rotated the mesh and generated 10 data samples. For each data sample, we compute a $512^3$ ground truth SDF volume using a CUDA-based program and render multi-view RGB and normal images using BlenderProc. In our experiments, the resolutions of the occupancy volume and sparse feature volume are 64 and 256, respectively. The resolution of the predicted and ground truth SDF volumes is 512. The model is trained with the Adam optimizer and a cosine learning rate scheduler. The loss weights $\lambda_1, \cdots, \lambda_6$ are set to 80, 2, 16, 2, 8, and 8, respectively.

All data preparation, including image rendering and SDF computation, is performed using an internal cluster. This process can be completed using 4000 CPU cores in roughly one week. The generated data takes up approximately 30TB. All model training tasks are conducted in public cloud clusters. Our main model is trained using 8 H100 GPUs for one week. All experiments listed in the paper can be completed in 15 days using 32 H100 GPUs (running multiple parallel experiments), excluding the preliminary exploration experiments.

**Architecture Details:** For VoxelFormer, the UNet consists of four levels with resolutions of $64^3$, $32^3$, $16^3$ and $16^3$. Each level includes a ResNet module, a projection-aware cross-attention module, and a downsampling module, with channel sizes of 64, 128, 256, and 512. We added 6 transformer layers at the bottleneck of the UNet, with each 3D voxel treated as a token, and token channels set to 512.

For SparseVoxelFormer, the sparse UNet consists of six levels with resolutions of $256^3$, $128^3$, $64^3$, $32^3$, $16^3$, and $16^3$. Each level includes a sparse ResNet module, a projection-aware cross-attention module, and a downsampling module, with channel sizes of 16, 32, 64, 128, 512, and 2,048. We added 16 transformer layers at the bottleneck of the UNet, with each 3D sparse voxel treated as a token, and token channels set to 1,024. The feature dimension of the output sparse feature volume (before the MLP) is 32.

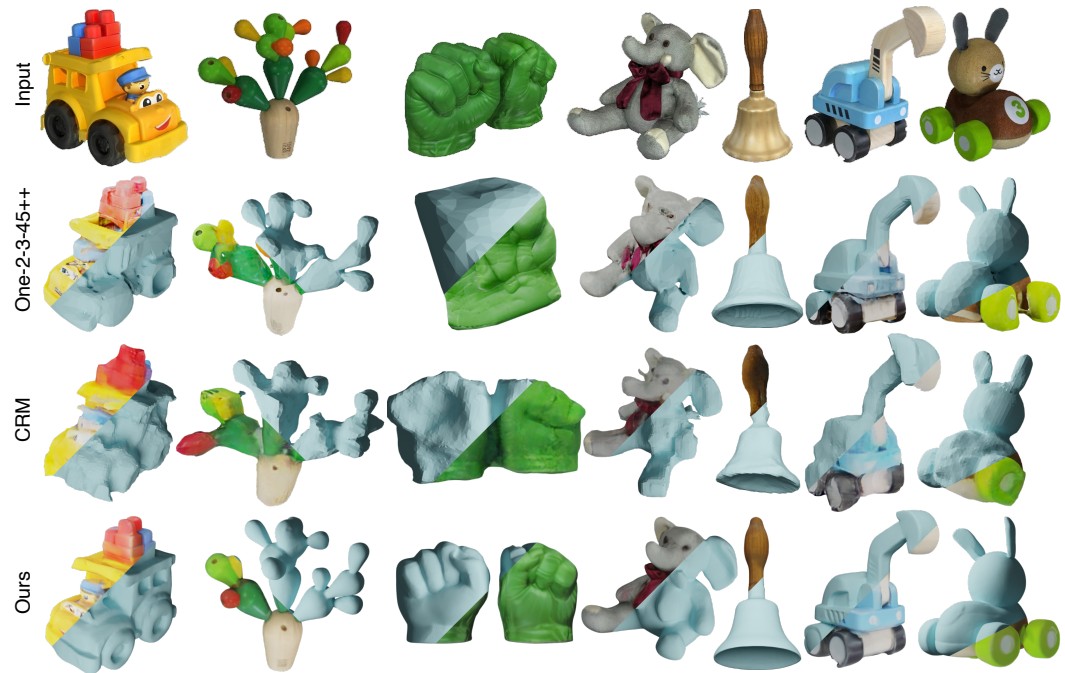

Figure 9: **Qualitative Results of One-2-3-45++ [31] and CRM [77] on Single Image to 3D.** Both the textured and textureless mesh renderings are shown.

For both of them, a skip connection is added to the UNet.

**Evaluation Metrics:** To account for the scale and pose ambiguity of the generated mesh from different baselines, we align the predicted mesh with the ground truth mesh prior to the evaluation metric calculation. This alignment process involves uniformly sampling rotations and scales for initialization and subsequently refining the alignment using the Iterative Closest Point (ICP) algorithm. We select the alignment that yields the highest inlier ratio. Both the ground truth and predicted meshes are then scaled to fit within a unit bounding box.

For 3D metrics, we sample 100,000 points on both the ground truth mesh and the predicted mesh and compute the F-score and Chamfer distance, setting the F-score threshold at 0.05. To evaluate texture quality, we compute the PSNR and LPIPS between images rendered from the reconstructed mesh and those of the ground truth. Following InstantMesh [85], we sample 24 camera poses, encompassing a full 360-degree view around the object, and utilize BlenderProc for rendering RGB and normal images with a resolution of 320×320. Since we use the VGG model for LPIPS loss calculation during training, we employ the Alex model for LPIPS loss calculation during evaluation.

### A.7 Training Details of MeshLRM

All results of MeshLRM, except those in Table 2, were reproduced by the MeshLRM authors at Hillbot following the original settings as described in the paper. For the results in Table 2, we trained the model using the same training data as our method on $8\times$H100 GPUs for 48 hours. We maintained the same batch size as reported in the paper and proportionally scaled down the original training time for each stage of MeshLRM based on a total training time of 48 hours. This included 5.8 seconds per iteration for 20,000 iterations in the 256-resolution pre-training, 12 seconds per iteration for 4,000 iterations in the 512-resolution fine-tuning, and 4.7 seconds per iteration for 4,000 iterations in mesh refinement.

### A.8 Qualitative Examples of One-2-3-45++ and CRM

Figure 9 shows qualitative results of One-2-3-45++ [31] and CRM [77] on single image to 3D and our method produces better results.

## A.9  Broader Impact

We introduce an efficient approach for training open-world sparse-view reconstruction models, which has the potential to significantly reduce energy consumption and carbon emissions, as baseline models typically require much more computing resources for training. Previously, the creation of 3D assets was reserved for specialized artists who spent hours or even days producing a single 3D model. Our proposed technique allows even novice individuals without specialized 3D modeling knowledge to create high-quality 3D assets in seconds. This democratization of 3D modeling has unleashed unprecedented creative potential and operational efficiency across various sectors.

However, like other generative AI models, it also carries the risk of misuse, such as spreading misinformation and creating pornography models. Therefore, it is crucial to implement strict ethical guidelines to mitigate these risks.

