# OpenReview forum: "MeshFormer : High-Quality Mesh Generation with 3D-Guided Reconstruction Model"
_NeurIPS.cc/2024/Conference — NeurIPS 2024 oral_

### Official Review · Reviewer_3423 · 2024-07-08

**Soundness:** 4
**Presentation:** 3
**Contribution:** 3
**Rating:** 7
**Confidence:** 4

**Summary:**

This paper introduces MeshFormer, a sparse-view reconstruction model designed to generate high-quality 3D textured meshes from sparse RGB images and their corresponding normal maps. By leveraging voxel representation, 3D inductive biases, SDF loss, and normal information, the model shows comparable inference performance to concurrent methods, while the entire training process can be completed using only 8 GPUs within a week (concurrent methods typically require around 100 GPUs). Experimental results demonstrate the effectiveness of the design.

**Strengths:**

1. The authors provided a detailed explanation of the motivations behind the model designs (including the introduction of voxel representation, the introduction of 3D full (or sparse) convolution, and so on) and demonstrated the reasonableness of these choices.

2. Compared to baseline methods, this model is simpler to train and demonstrates better qualitative and quantitative results.

3. The ablation study demonstrates the effectiveness of normal input, SDF supervision, geometry enhancement, and other methods proposed in the paper.

**Weaknesses:**

1. Although the authors provide detailed textual descriptions in the method section, it would be better if more mathematical symbols and equations were used, which could explain the entire pipeline more clearly and unambiguously.

2. For reproducibility, the authors should provide more implementation details, including a more detailed model architecture, the values of hyperparameters (e.g., \lambda in the loss function), and other relevant information.

3. The authors don’t report the comparison of inference time and memory usage between the proposed model and the baseline models.

**Questions:**

1. Can the normal maps of the mesh be completely consistent with the normal maps predicted by the model after the post-processing algorithm?

**Limitations:**

Yes, the authors addressed limitations, potential negative societal impact, and mitigation.

---

> ### Author Rebuttal · Authors · 2024-08-07
>
> ## More mathematical symbols and equations
> Thank you for pointing this out. We will follow your suggestion to include more mathematical symbols and equations in our revision when introducing the method.
>
> ## More implementation details
> We will follow the suggestion to include more implementation details in our revision, such as the specifics of the network architecture and the weights of the loss terms.
>
> ## Inference time and memory usage
> We followed your suggestion to include a comparison of inference time and memory usage in the rebuttal PDF (please refer to Table 3). We found that while our method generates meshes at the highest resolution (512^3, 8x larger than 256^3), we still maintain a competitive speed and memory footprint.
>
>
> ## Normal maps of the mesh (after post-processing) vs. normal maps predicted by the model
> Unfortunately, the post-processing we used cannot guarantee that the mesh normals will be completely aligned with the predicted normal maps after processing. This is because the algorithm operates in local space and avoids large vertex movements. Additionally, the predicted normal maps may contain errors or conflicts, such as inconsistent neighboring normals, which cannot be perfectly matched. The adopted algorithm is an iterative numerical optimization method and does not compute an analytic solution.
>
> However, we have quantitatively verified that the post-processing module can significantly improve mesh normal consistency with the predicted normal map. For example, before post-processing, only 26.4% of mesh vertices had a normal angle error of less than 2 degrees. After post-processing, this number increased to 40.8%. For a 10-degree threshold, the ratio increases from 78.8% to 86.4%. For more details, please refer to Table 4 in the rebuttal PDF.
>
> We have also included qualitative examples to illustrate the importance of this post-processing module in recovering sharp geometric details and reducing noisy artifacts induced by the marching cubes algorithm. Please check out Figure 5 of the original paper.

---

> > ### Comment · Reviewer_3423 · 2024-08-13
> > **Comments on Rebuttal**
> >
> > I am glad the authors agreed to include these discussions in the final revision and provided more experimental results in rebuttal. I will keep my positive score.

---

### Official Review · Reviewer_bHQc · 2024-07-08

**Soundness:** 3
**Presentation:** 3
**Contribution:** 3
**Rating:** 8
**Confidence:** 5

**Summary:**

The paper proposes a high-quality feed-forward 3D object reconstruction method from sparse view RGB images. It uses an explicit voxel structure for better geometric inductive bias, auxiliary inputs such as 2D diffusion generated normal images and SDF representation for better geometric details, and an end-to-end trainable pipeline that eliminates the need for multi-stage refinement. The method gives high quality reconstruction results, especially in terms of fine-grained and smooth geometry.

**Strengths:**

1. Although the network architecture and 3D representations are more complicated than previous methods, they are end-to-end differentiable and alleviate the training burden of multi-stage refinement.
2. The idea of using 2D diffusion generated normal images as input to the reconstruction pipeline is interesting and insightful.
3. It is more computationally efficient to train (Line 73).
4. The qualitative results are impressive, especially the mesh normals.

**Weaknesses:**

1. In original LRM the only supervision signal needed is RGB images. The proposed method, however, needs access to the full 3D shape for supervising the occupancy. It is fine for hand-made 3D assets but might poses some difficulty when trying to scale to real datasets.

**Questions:**

1. Table 3 row (a) shows the impact of normal input. When you remove the normal input, do you also remove the normal output and the normal loss? I ask this because in section 3.3 you say learning from RGB to geometric details directly can be difficult, so it makes more sense to just remove the normal input but preserve normal supervision to compare.

**Limitations:**

1. It requires 2D diffusion models to generate auxiliary inputs, which can drastically slow down the reconstruction speed.

---

> ### Author Rebuttal · Authors · 2024-08-07
>
> ## Geometry supervision for real-world training datasets
> We agree that image supervision is easier to add when extending to real-world training datasets. However, it is not impossible to obtain corresponding depth maps and even meshes for real-world RGB images, such as through depth sensors or Structure from Motion (SfM). Given the depth map (and even meshes), we can still apply direct 3D geometry supervision. If full 3D shapes are not captured, we can also apply partial supervision to the visible views (only supervising the visible points) while generating the full shape. We acknowledge that this may require more advanced techniques and designs, and we leave it as a promising avenue for future work.
>
> ## Table 3, row (a):
> Yes, for Table 3, row (a) of the paper, we remove the normal input but preserve the normal output and normal supervision.
>
> ## 2D diffusion models drastically slow down the reconstruction speed
> We currently use the normal ControlNet of Zero123++ to generate the multi-view normal inputs. It tiles the multi-view RGB images as a single input condition and generates the tiled normal maps in a single diffusion process, which takes approximately 4.1 seconds on an H100 GPU. For the application of single-image to 3D, generating tiled multi-view RGB images takes about 3.6 seconds. The total time on the 2D diffusion model side is only around 7.7 seconds, which is still acceptable.

---

> > ### Comment · Reviewer_bHQc · 2024-08-14
> >
> > Thanks for the rebuttal. I keep my original rating.

---

### Official Review · Reviewer_mWQL · 2024-07-10

**Soundness:** 3
**Presentation:** 4
**Contribution:** 3
**Rating:** 7
**Confidence:** 5

**Summary:**

In this work, the authors propose a sparse view reconstruction model that utilizes a set of images (with camera poses) and corresponding normal maps to produce a reconstructed textured mesh. The primary contribution lies in adopting voxel-based 3D representation and employing a network architecture that integrates both 3D convolution and attention layers. Moreover, direct geometry supervision (SDF loss) is applied during the training process, alongside rendering-based losses. Experimental results demonstrate that the generated 3D shapes achieve state-of-the-art performance when compared to existing works on the single-view to 3D task.

However, as highlighted in the weakness section, there are potential misclaims regarding the technical contributions. It is highly recommended to revise the manuscript to cite and discuss these related works. Despite this, I am currently inclined towards accepting the paper and would be happy to champion it if the aforementioned issues are addressed in the final version.

**Strengths:**

- The writing is clear and easy to follow.
- The combination of SDF loss and rendering losses appears novel for training a feed-forward based network. Additionally, the ablation study in Table 3(b) clearly indicates that SDF supervision is crucial for achieving good geometry, as evidenced by the significant CD difference between (b) and (g).
- Although [33] has explored using normal maps for the reconstruction task, it seems new to employ normal maps as inputs and supervision for a feed-forward reconstruction network.
- Experimental results demonstrate state-of-the-art performance over existing baselines, as shown in Table 1 and Figure 3. Furthermore, it is illustrated that existing methods cannot achieve similar performance given the same computational resources (Table 2).
- The ablation study confirms that various components are essential for the final performance, including considering normal input and SDF supervision.

**Weaknesses:**

Possibly Misclaimed Technical Novelties:

However, the current manuscript may contain several misclaims regarding its technical novelties.

One claimed novelty is the adoption of a 3D voxel representation. However, the use of 3D voxel-like volumes in reconstruction is not a new idea and has been well-explored in various works, including:

A. Generalized Deep 3D Shape Prior via Part-Discretized Diffusion Process, CVPR 2023

B. SDFusion: Multimodal 3D Shape Completion, Reconstruction, and Generation, CVPR 2023

C. Locally Attentional SDF Diffusion for Controllable 3D Shape Generation, SIGGRAPH 2023

D. One-2-3-45++: Fast Single Image to 3D Objects with Consistent Multi-View Generation and 3D Diffusion, CVPR 2024

E. Make-A-Shape: a Ten-Million-scale 3D Shape Model, ICML 2024

Additionally, the use of convolution + transformer layers to process grid input seems to be standard procedure in 2D generation tasks, as seen in:

Diffusion Models Beat GANs on Image Synthesis, NeurIPS 2021

Similar architectures have also been widely adopted in some of the aforementioned 3D reconstruction works, such as [A, C, D, E].

Regarding image conditioning, the cross-attention with image patch features is also well-explored in various works mentioned above, such as [C, D, E].

**Questions:**

Some suggestions:
- Considering the above existing and concurrent works (Weakness Section), it is difficult to be convinced that some of the proposed modules are novel. It is highly recommended to cite and discuss the differences with these prior works and adjust the claims accordingly.
- Although it is acknowledged in the limitation section that the reconstruction performance will be affected by the errors of 2D models, it is recommended to include this as one of ablation case in Table 3 to better visualize this limitation.
- Furthermore, as no real-world images have been tested within the proposed framework, it is advisable to avoid from using the term "open-world" (L384) to describe the current framework in order to prevent overclaims.

**Limitations:**

The main limitation is well described in Section 5.

---

> ### Author Rebuttal · Authors · 2024-08-07
>
> # No real-world images tested?
> We would like to clarify that one of our main testing datasets, OmniObject3D, is a real-world scanned 3D dataset. In addition, we also include some qualitative examples with real-world input in our rebuttal PDF (see Fig. 3), where MeshFormer performs quite well.
>
> The term 'open-world' means that MeshFormer differs from many previous methods (such as A, B, C listed by the reviewer). Those methods are trained on datasets with a limited number of object categories (e.g., tens of categories in ShapeNet) and cannot generalize to novel categories. Unlike those methods (e.g., 3D native diffusion), MeshFormer takes as input sparse-view images and normal maps generated by 2D diffusion models, and demonstrates much stronger generalizability. MeshFormer is thus not limited to the training 3D dataset and can handle arbitrary object categories.
>
> # Ablation study of 2D model errors
> We would like to clarify that, in Tab. 3 of the paper, we have analyzed the influence of errors from 2D normal models (rows f and g). Additionally, we provide some qualitative examples in Fig. 8.
>
> For the effect of multi-view RGB, please compare Tab. 1, row ‘Ours’ (predicted RGB) and Tab. 3, row f (ground truth RGB) of the paper.
>
> # Claims about technical novelties
> Thank you for pointing this out. We will cite these prior works and discuss the differences in our revision. We fully understand your concerns and would like to address the potential misunderstanding in detail.
>
> ## Point 1
>
>  The reviewer summarized that the "primary contribution lies in adopting a voxel-based 3D representation and employing a network architecture …." **We respectfully disagree with this argument.** Our main claim is that by proposing a 3D-guided reconstruction model that explicitly leverages 3D native structure, input guidance, and training supervision, MeshFormer can significantly improve both mesh quality and training efficiency. Our main findings/contributions include:
>
> - (a) Using normal images as additional input in feed-forward reconstruction models greatly enhances the prediction of geometric details.
>
> - (b) Proposing to learn and output a 3D normal map, which can be used for further geometric detail enhancement.
>
> - (c) Combining SDF loss and rendering losses in training feed-forward reconstruction models enables a unified single-stage training process. In contrast, concurrent works rely on complex multi-stage "NeRF-to-Mesh" training strategies to export high-quality meshes (e.g., MeshLRM, InstantMesh) and struggle to generate high-quality geometry.
>
> - (d) Explicitly leveraging 3D native voxel representation, network architecture, and projective priors fosters faster convergence speeds and significantly reduces the training requirement.
>
> **We would like to emphasize that all four points are crucial to MeshFormer.  We thus do not agree that our primary contribution lies solely or primarily in adopting 3D representation and network architecture.** For example, without points (a), (b), and (c), our mesh quality would be significantly compromised.
>
> ## Point 2
> We totally agree that the utilization of 3D voxel representation is common in general 3D generation. However, **all works listed by the reviewer (A-E) focus on 3D native diffusion, one of the paradigms in 3D generation, which differs from the route of MeshFormer.** There are some common limitations of this line of work. For instance, all of A-E focus on geometry generation only and cannot predict high-quality texture directly from the network. Also, due to the limited amount of 3D data, 3D native diffusion methods typically struggle with open-world capability and focus on closed-domain datasets (e.g., ShapeNet) in their experiments (A, B, C).
>
> In MeshFormer, **we aim to achieve direct high-quality texture generation and handle arbitrary object categories**. We are thus following another route: sparse-view feed-forward reconstruction, instead of the 3D native diffusion. In this specific task setting, **many of the works provided by the reviewer are not suitable for comparison in our experiments**. More comparable works are recent LRM-style methods (e.g., InstantMesh, MeshLRM, LGM, TripoSR, etc). However, **most of them only utilize the combination of triplane representation and large-scale transformers**.
>
> In our paper, we do not claim to be the first to use voxel representation in 3D generation. Instead, we would like to share our findings:
>
> - In this specific task setting (open-world sparse-view reconstruction with feed-forward textures), we are not limited to the triplane representation. 3D native structures (voxels), network architectures, and projective priors can facilitate more efficient training and significantly reduce the training resources required (from over one hundred GPUs to only 8 GPUs).
> - In this specific task setting, we require a scalable network to learn a lot of priors. However, not only triplane-based transformers can be scalable. When marrying the 3D convolution with transformer layers, it can also be scalable.
> - In addition to using 3D native representation and networks in 3D native diffusion, we can also combine them with differentiable rendering to train a feed-forward sparse-view reconstruction model with rendering losses.
> - For image conditioning, C and E only take a single image as a condition. D **first employs max pooling across multi-view features** and then uses cross-attention across the pooled multi-view features and the voxel feature. The max pooling is typically affected by occlusion (voxels are not visible in all views) and thus becomes less effective. Instead, we propose using cross-attention across all multi-view projected image features and voxel features, which can implicitly leverage structure and visibility priors to focus only on visible regions. We demonstrate that this strategy is more efficient than the pooling strategies used in D as shown in Tab. 3 of the paper and Tab. 1 of the rebuttal PDF.

---

> > ### Comment · Reviewer_mWQL · 2024-08-12
> > **Comments on Rebuttal**
> >
> > Thanks for the additional experiments and comments. They have addressed my previous concerns.
> >
> > Regarding the technical contribution, it is understood that the authors would like to highlight the contributions of applying a voxel representation and corresponding networks instead of a tri-plane representation in an open-world reconstruction task. I agree that this work has shown good evidence for the necessity of this choice, and this should be recognized.
> >
> > Despite this, I believe that a comprehensive discussion of related works, regardless of in-categories setting ([A-C]) or open-world setting (D-E), would help readers understand the field's development. I am glad the authors agreed to include these discussions in the final revision and adjust my rating to acceptance.

---

> > > ### Author Response · Authors · 2024-08-13
> > > **Thank you**
> > >
> > > Thank you for adjusting the score! We will follow your suggestion to cite the mentioned works and include a comprehensive discussion.

---

### Official Review · Reviewer_V11k · 2024-07-17

**Soundness:** 4
**Presentation:** 4
**Contribution:** 3
**Rating:** 7
**Confidence:** 5

**Summary:**

This paper proposes an improved framework for feed-forward reconstruction models. The authors advocate a number of improvements over the initial design of Large Reconstruction Model, including model architecture and training schemes. Experiments show that the method reconstructs better geometry and texture on Google Scanned Objects and OmniObject3D datasets.

**Strengths:**

- The paper is focused on ablating different components for feed-forward sparse-view reconstruction, and in-depth analyses are provided for each design choice. Although there are no complicated new method proposed, such analysis bring value for understanding how and why each component works.
- The proposed method is evaluated on (preprocessed) real-world multi-view datasets, showing improvements over baselines on all metrics. Extensive ablative analyses are also provided to better understand the behaviors of the proposed method.

**Weaknesses:**

- Since this is more of an analysis paper, it would be good if the authors could also document the other components that were tried/ablated but did not see significant differences.
- Since training resources was discussed and compared, it would be nice if there could be an analysis on the mesh generation/reconstruction quality over training time.

**Questions:**

Please see the questions in the weakness section.

**Limitations:**

Yes

---

> ### Author Rebuttal · Authors · 2024-08-07
>
> ## Experiments tried/ablated but did not show significant differences
> We are happy to follow the reviewer's suggestions to include more discussions about the experiments we have conducted in our revision, such as:
> - the difference between joint training and separate training of the dense model and sparse model;
> - the difference between max-pooling, mean-pooling, and cross-attention in projection-aware feature aggregation;
> - the comparison of inference time and memory consumption with baseline methods;
> - mesh generation quality over training time;
> - quantitative analysis of the geometry enhancement module;
> - more qualitative examples on real-world images.
>
> Please let us know if there are any additional experiments you are interested in.
>
>
> ## Mesh generation/reconstruction quality over training time
> Our MeshFormer can be trained efficiently with only 8 GPUs, generally converging in roughly two days. We have followed the suggestion to include a quantitative analysis of our mesh generation quality over training time. As shown in Table 2 of the rebuttal PDF, the performance quickly improves and nearly converges with marginal changes after the two-day training period.

---

### Official Review · Reviewer_r7MY · 2024-07-17

**Soundness:** 3
**Presentation:** 3
**Contribution:** 3
**Rating:** 7
**Confidence:** 5

**Summary:**

In this work, the authors propose MeshFormer, a sparse-view reconstruction model that explicitly leverages 3D native structure, input guidance, and training supervision. They leverage 3D sparse voxels as their representation and combine transformers with 3D (sparse) convolutions to inject 3D prior. Additionally, they propose to take the corresponding normal maps together with sparse-view RGBs as input and also generate them as output, which could be used for geometry enhancement. Extensive experiments show that MeshFormer can be trained efficiently and outperforms state-of-the-art methods in terms of generating high-quality textured meshes.

**Strengths:**

- MeshFormer is able to generate high-quality textured meshes with fine-grained geometric details.

- The authors find that using normal images together with RGB images greatly helps in predicting geometric details. Additionally, the model outputs a normal map, which can be used for geometry enhancement.

- The proposed method explicitly leverages 3D native structure, input guidance, and training supervision, resulting in faster convergence speed and better geometric details.

**Weaknesses:**

- Pixel-based 2D methods (e.g., LGM) can preserve thin details, while 3D-based methods often smooth these details. How do you justify that? For example, in Figure 3 Column 4, the loose thread of the toy is captured by LGM, while MeshFormer ignores it.

- The proposed name "VoxelFormer" seems improper to me. It seems more like a 3D UNet with a deep bottleneck composed of multiple transformer layers.

- The projection-aware cross-attention layer projects 3D voxels onto the m views to interpolate m RGB and normal features. However, in the object case, one 3D voxel usually only corresponds to one view (due to occlusion). This cross-attention is projection-aware but not truly 3D-aware. Have you tried some occlusion-aware attention in your sparse model? Since you already have the coarse structure of the object, it could be used to filter out unneeded features.

- According to Table 3 (d), you mention "we replace the cross-attention with simple average pooling and observe a significant performance drop." Could you also try max-pooling? Additionally, do you concatenate the 3D feature voxel at every level of the network, as done in One-2-3-45++?

**Questions:**

- Do you use a shared backbone (trainable DINOv2) for both RGB and normal images? Do you use Plücker embedding here?

- Could you provide a more detailed description for the Sparse VoxelFormer architecture? For example, how many sparse convolution layers are used in each resolution?

- Instead of joint training, have you tried splitting the dense model and sparse model for two-stage training?

- The output voxel resolution is $256^3$, while the SDF supervision is $512^3$. I notice that there is an interpolation step in Figure 2. It would be better to add a short text description for this.

- Do you use GT multi-view normals for the teaser? If you use the GT normal images, please include that in the caption.

- I suggest discussing XCube [a] in your literature review. XCube also utilizes sparse voxels as their 3D representation and leverages 3D sparse UNet with transformer layers. Additionally, they generate 3D shapes in a coarse-to-fine manner and use tiny MLPs to predict various attributes, such as normals, semantics, and SDF.

[a] XCube: Large-Scale 3D Generative Modeling using Sparse Voxel Hierarchies. CVPR 2024.

**Limitations:**

The authors already include limitations and broader impact in the paper.

---

> ### Author Rebuttal · Authors · 2024-08-07
>
> ## Thin structures
> We would like to clarify that the loose thread of the toy was not displayed due to a slight pose mismatch when visualizing the results. In fact, the loose thread is reconstructed by our MeshFormer. We have included additional views of our generated results (see Figure 1 of the rebuttal PDF), where the loose thread can be observed. You can check the video on our website.
>
> MeshFormer generates high-resolution (512) SDF volumes, which are sufficient to preserve very thin structures, as shown in Figure 2 of the rebuttal PDF.
>
> ## The name "VoxelFormer"
> The meaning of "VoxelFormer" is two-fold. On the one hand, it refers to combining voxel representation and transformer layers, in contrast to recent LRM-based methods that rely on triplane representation and transformers to achieve scalability. On the other hand, it can also be interpreted as Voxel-Form-er, meaning the module that generates the voxels. We are open and happy to consider other names if the reviewer has more suitable suggestions.
>
> ## Occlusion-aware feature aggregation
> We agree that occlusion-aware feature aggregation is very important. This is the primary reason we use a cross-attention layer to aggregate the projected multi-view features instead of using a simple average or max pooling method like previous approaches. We hope the cross-attention layer can implicitly utilize prior knowledge of coarse structure and visibility to focus on the visible views. Our experiments also verify its superiority over mean pooling aggregation (Table 3 (d) of the paper). While we could explicitly filter out some views according to the predicted occupancy from the first stage, as the reviewer suggested, we would like to point out that the occupancy band has some thickness, and accurately determining the visibility of each voxel can be quite challenging. This may require more advanced techniques, and we leave this as a promising future direction.
>
> ## Table 3 (d)
> We followed the suggestion to add an ablation variant using the max-pooling aggregation mechanism (please refer to Table 1 of the rebuttal PDF). We found that while the max-pooling aggregation performs slightly better than average pooling, it is still significantly inferior to our projection-aware cross-attention mechanism.
>
> Yes, we follow the "skip-connection" scheme of the typical UNet to concatenate the voxel features before the bottleneck with the voxel features after the bottleneck. If this is not what you are asking, please let us know.
>
> ## Shared backbone and Plücker embedding
> Yes, we use a shared backbone (trainable DINOv2) for both RGB and normal images. We did not use Plücker embedding in our experiments. Instead, we leveraged the camera poses to unproject 2D image features into 3D feature volumes.
>
> ## Detailed description of the sparse VoxelFormer architecture
> Yes, we will follow your suggestion to include more details about the Sparse VoxelFormer architecture in our revision, such as the number of sparse convolution layers used at each resolution.
>
> ## Joint training
> Yes, we began our experiments by training the dense model and sparse model separately. However, we found that this approach leads to a domain gap for the occupancy field during inference, as the predicted occupancy by the dense model may be imperfect, while the sparse model is only trained with ground truth occupancy. Joint training can mitigate this gap and reduce artifacts during inference.
>
> ## Interpolation from $256^3$ to $512^3$
> Thank you for pointing this out. We will add the missing description in our revision. Specifically, we generate a sparse feature volume with a resolution of 256, and then trilinearly interpolate it to a sparse feature volume with a resolution of 512. This interpolated sparse features are then fed into the SDF decoder for predicting SDF values, which are subsequently used to compute the loss against the 512-resolution ground truth SDF.
>
> ## Multi-view normals for the teaser
> Yes, we use GT multi-view normals for the teaser. We will make this clearer in our revision.
>
> ## XCube
> Thank you for pointing this out. We will cite XCube and discuss it in our revision. We agree that XCube shares many high-level ideas with us, such as hierarchical sparse voxel representation and coarse-to-fine generation. However, we would like to point out that they follow the paradigm of the 3D native diffusion, which can only generate geometry but fails to directly predict texture from the model. In contrast, we follow the paradigm of the feed-forward sparse-view reconstruction and incorporate differentiable rendering into the pipeline, which enables the network to directly generate high-quality texture. Additionally, we combine 3D (sparse) convolution with the transformer layer to increase capacity and scalability of the network, while they mainly rely on 3D convolution only and may be limited in both capacity and scalability.

---

> > ### Comment · Reviewer_r7MY · 2024-08-14
> >
> > Thanks for the rebuttal. I would like to accept this paper. Please include the promised changes in the revision.

---

### Author Rebuttal · Authors · 2024-08-07

We thank all reviewers for their insightful comments and valuable suggestions. We are pleased to note that all five reviewers were supportive of our work:

- They complimented the impressive mesh quality with fine-grained geometric details (r7MY, bHQc, 3423, mWQL, V11k).
- They praised our fast training speed and significantly reduced computational resources (r7MY, bHQc, 3423, mWQL).
- They noted that our method is well-motivated (3423) and the paper is well-written (mWQL).
- They highlighted our qualitative state-of-the-art performance (r7MY, mWQL, V11k, 3423) and insightful and extensive ablation study (mWQL, V11k, 3423).
- They acknowledged the novelty and benefits of our concrete findings/contributions:
  - Using normal images as additional input for feed-forward models greatly helps predict geometric details. (r7MY, mWQL, bHQc, 3423).
  - Proposing to output a normal map, which can be used for geometry enhancement (r7MY, 3423).
  - The combination of SDF loss and rendering losses enables unified single-stage training and achieves good geometry (r7MY, mWQL, bHQc, 3423).
  - Explicitly leveraging 3D native structures and projective priors fosters faster convergence speed (r7MY, bHQc, 3423).

We have also included a PDF with some figures and tables to address the specific concerns raised by the reviewers.

---

### Decision · Program_Chairs · 2024-09-25

**Decision:**

Accept (oral)

**Comment:**

Post-rebuttal, the paper is a clear accept with all five reviewers scoring the paper as an Accept or higher. Reviewers praised the performance, novelty, and efficiency of the proposed end2end design.

In their rebuttal, authors have promised to include certain additional analyses, such as:
- max-pooling vs. attention-based multi-view attention ablation (r7MY)
- additional analysis on which ideas did not work (V11k)
- the performance of the model as a function of the training time (V11k)
- better exposition using math with more implementation details (3423)
- speed & memory consumption (3423).
Authors are strongly encouraged to inlude the latter in the final camera-ready version.